# CNCA: Toward Customizable and Natural Generation of Adversarial Camouflage for Vehicle Detectors

**Linye Lyu, Jiawei Zhou, Daojing He, Yu Li**
Harbin Institute of Technology, Shenzhen
{lyulinye, zhoujiawei6666, yu.li.sallylee}@gmail.com
hedaojinghit@163.com

## Abstract

Prior works on physical adversarial camouflage against vehicle detectors mainly focus on the effectiveness and robustness of the attack. The current most successful methods optimize 3D vehicle texture at a pixel level. However, this results in conspicuous and attention-grabbing patterns in the generated camouflage, which humans can easily identify. To address this issue, we propose a Customizable and Natural Camouflage Attack (CNCA) method by leveraging an off-the-shelf pre-trained diffusion model. By sampling the optimal texture image from the diffusion model with a user-specific text prompt, our method can generate natural and customizable adversarial camouflage while maintaining high attack performance. With extensive experiments on the digital and physical worlds and user studies, the results demonstrate that our proposed method can generate significantly more natural-looking camouflage than the state-of-the-art baselines while achieving competitive attack performance. Our code is available at https://github.com/SeRAlab/CNCA.

## 1 Introduction

Over the past years, Deep Neural Networks (DNNs) have revolutionized a wide range of research domains, especially in computer vision tasks, such as image classification, object detection, and semantic segmentation. DNNs are widely used in real-world systems, such as face recognition and autonomous driving. Despite their impressive success, DNNs are found vulnerable to adversarial examples [20], which are carefully crafted to deceive DNNs.

Generally, adversarial attacks can be classified into two categories: digital attacks, which primarily add small pixel-level perturbations to the input images; physical attacks, which manipulate the object's physical properties, such as its shape, surface, or surroundings, to deceive the target model in the real world. Physical attacks are more challenging than digital attacks, as they must remain effective under various complex physical conditions, including different viewing angles, distances, and lighting conditions. This paper focuses on physical attacks against vehicle detection models since they play critical roles in real-world applications like surveillance and autonomous driving systems.

38th Conference on Neural Information Processing Systems (NeurIPS 2024).

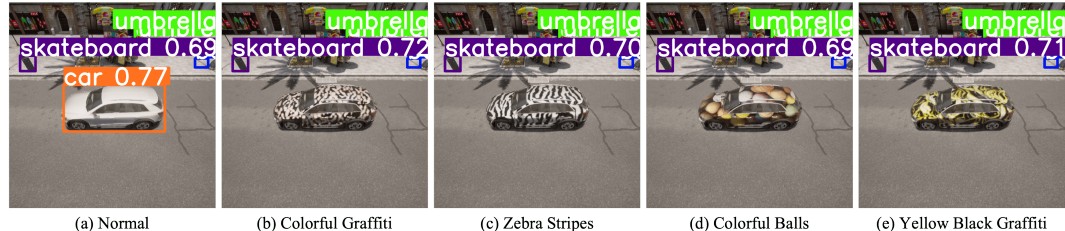

| (a) Normal | (b) Colorful Graffiti | (c) Zebra Stripes | (d) Colorful Balls | (e) Yellow Black Graffiti |

Figure 1: Customized and natural adversarial camouflage with various styles. (a) A car with normal texture; (b)(c)(d)(e) are the different styles of camouflage generated by our method CNCA. Their captions are user-specified input prompts.

Prior advanced physical attacks against vehicle detectors use adversarial camouflage technique [22; 21; 18; 19]. This technique fully covers the whole vehicle's surface with adversarial texture, which leads to better attack performance regardless of the viewing positions. Leveraging a differentiable renderer can effectively optimize 3D vehicle texture to deceive vehicle detectors via gradient back-propagation. However, all these methods suffer certain issues. Firstly, there is no prior knowledge of naturalness to guide the camouflage generation, resulting in conspicuous and attention-grabbing camouflage patterns as shown in Table 4. Secondly, all the current methods optimize the adversarial camouflage at a pixel level, making it challenging to resemble natural-looking patterns. Last but not least, none of the methods can customize the appearance of camouflage, making it hard to adapt to specific environments like forests and deserts.

To address the above issues, we propose CNCA, a novel framework to generate customizable and natural adversarial camouflage against vehicle detectors as shown in Figure 1. Our insight is that: to gain naturalness and customizability, we need to leverage models that are equipped with prior knowledge of naturalness and allow conditional input signals. Motivated by this insight, we leverage an off-the-shelf pre-trained diffusion model to generate adversarial texture images with user-specific text prompts. The challenge of this approach is how to guide the adversarial gradient from the detection model to the image generation process. We introduce an adversarial feature to combine with the original text prompt feature. The combined feature forms the conditional input of the diffusion model. Thus, the resulting image is both natural and adversarial. Furthermore, we apply a clip strategy to the adversarial features to balance the trade-off between naturalness and attack performance. The combination of diffusion models, adversarial features, and clipping strategies facilitates the generation of customizable and natural camouflage.

The main contributions of our work are summarized as follows:

- To the best of our knowledge, our work is the first to investigate natural physical adversarial camouflage generation with diffusion models. It is also the first that can generate various styles of adversarial camouflage against vehicle detectors.
- We introduce an adversarial feature that can be combined with the conditional input of the diffusion models, enabling gradient-based adversarial camouflage generation.
- We propose to apply a clipping strategy for the adversarial feature to balance the trade-off between naturalness and attack performance.

We conduct a comprehensive evaluation with popular vehicle detectors and datasets in both digital and physical settings, and the results show that our method is effective in generating natural and customized adversarial camouflage.

## 2 Related Work

**Adversarial Camouflage.** Accurate detection of nearby vehicles is a crucial safety requirement of self-driving cars. Therefore, there has been a growing interest in crafting adversarial camouflage to attack vehicle detection systems. Most current research uses a 3D simulation environment [3] to generate 2D rendered vehicle images with various transformations to develop robust adversarial camouflage. Early works of adversarial camouflage against vehicle detection are mostly black-box because the rendering process of the traditional rendering method is non-differentiable. The first

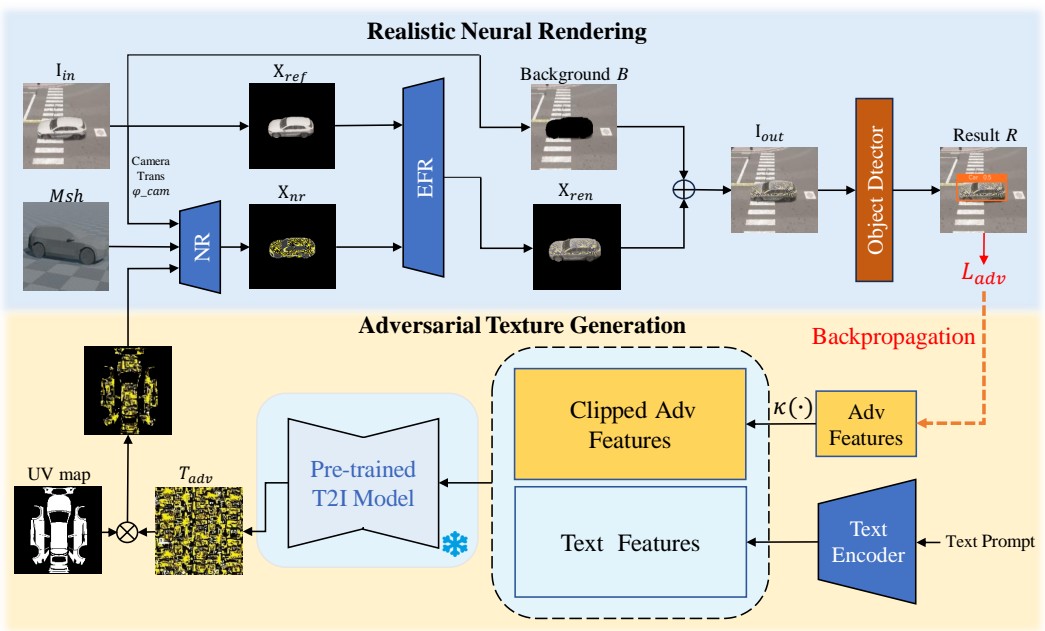

Figure 2: CNCA framework for generating customizable and natural adversarial camouflage.

work of vehicle adversarial camouflage [25] CAMOU trains a neural network to mimic the behavior of both the rendering and detection of the camouflage vehicles. Then, they can use this network to optimize the adversarial texture. [23] propose to use a genetic algorithm to search the optimal parameters for the synthesis of the adversarial texture pattern. Then, they enlarge and repeat the pattern to cover the whole surface of the vehicle.

Recent advanced methods introduce neural rendering to enable direct optimization of adversarial texture via gradient back-propagation algorithms. Dual Attention Suppression attack (DAS) [22] suppresses model and human attention on the camouflaged vehicle. However, it suffers a limited attack success rate because the adversarial pattern only covers part of the vehicle surface. Then, Full-Coverage Attack (FCA) [21] optimizes the entire surface of the vehicle in multi-view settings. Furthermore, Differentiable Transformer Attack (DTA) [18] proposes a differentiable renderer that can express complex environment characteristics like shadow and fog on the vehicle surface. ACTIVE [19] introduces a new texture mapping that incorporates depth images and tries to improve the naturalness of the camouflage by using larger texture resolutions and applying a smooth loss. Despite prior works achieving impressive attack performance, these methods optimize the camouflage patterns at the pixel level without prior knowledge of naturalness. Consequently, the generated camouflage is conspicuous and attention-grabbing for human observers.

**Diffusion Models.** Diffusion Models (DMs) [6] are widely used to generate natural images of higher quality and diversity. Since billions of image-text dataset pairs [16] are used to train these models, DMs provide a strong prior knowledge of natural and realistic images and their corresponding text captions. As a result, DMs can produce highly realistic and varied images across different user-specific prompts.

## 3 Methods

In this section, we present an overview of our framework for generating customizable and natural adversarial camouflage while maintaining comparable attack performance. Subsequently, we provide a detailed explanation of the essential components of our framework.

### 3.1 Overview

Figure 2 illustrates our whole framework for adversarial camouflage generation. First, we obtain a vehicle image dataset from the Carla simulation environment. The dataset includes the original input

images $I_{in}$, ground truth labels $Y$, camera pose parameters $\Phi_{cam}$ (position and angle), and vehicle mask $M$. With $I_{in}$ and $M$, we can obtain the background images $B$ and foreground vehicle reference images $X_{ref}$:

$$B = I_{in} \cdot (1 - M) \tag{1}$$
$$X_{ref} = I_{in} \cdot M \tag{2}$$

Then, we use a neural renderer $NR$ to obtain rendered vehicle images $X_{nr}$ with 3D mesh $Msh$, UV texture image obtained from UV map mask and adversarial texture image $T_{adv}$, and camera parameters $\Phi_{cam}$ of the vehicle. Next, $X_{ref}$ and $X_{nr}$ forward into a neural network called Environment Feature Renderer (EFR), which extracts the environmental features from $X_{ref}$ and render these features into $X_{nr}$ to obtain $X_{ren}$. We then add $X_{ren}$ with background $B$ to obtain the realistic camouflaged vehicle images $I_{out}$. Then, we input $I_{out}$ into the target object detector to obtain the detection results $R$.

Since we introduce a pre-trained text-to-image (T2I) diffusion model to generate the UV-map texture images, we can provide text prompt $P_{txt}$ to customize the texture image. The text encoder processes the text prompt to obtain the text feature $F_{txt}$. Then, the text feature is combined with the adversarial feature $F_{adv}$, which will be optimized during the camouflage generation. We apply a clip function $\kappa(\cdot)$ to $F_{adv}$ during optimization to balance naturalness and attack performance. Then, the combined features are fed into the pre-trained T2I model, which outputs the adversarial texture images $T_{adv}$.

$$F_{txt} = enc(P_{txt}) \tag{3}$$
$$T_{adv} = T2I([\kappa(F_{adv}), F_{txt}]) \tag{4}$$

In the end, we can obtain the final adversarial camouflage by minimizing the below adversarial loss function from the target detector:

$$D_s(x) = \text{IoU}\left(D_b(x), gt\right) \cdot D_c(x) \cdot D_o(x)$$
$$L_{adv}(x) = -\log\left(1 - \max\left(D_s(x)\right)\right), \tag{5}$$

where $x$ is the input image for the target detector, $D_b(x)$ is the detection bounding box, $gt$ is the ground-truth bounding box. We calculate the Intersection over Union (IoU) between $D_b(x)$ and $gt$. This IoU score allows the optimization to focus on the bounding box with larger intersections with ground truth. $D_o(x)$ and $D_c(x)$ are the objectiveness score and the class confidence score for the bounding box, respectively. We obtain our detection score $D_s(x)$ by multiplying the IoU score, objectiveness score, and class confidence score. We select the highest $D_s(x)$ to compute $L_{adv}(x)$ using a log loss. By minimizing $L_{adv}(x)$, we encourage the camouflaged vehicle to be undetected or misclassified by the detector.

## 3.2 Realistic Neural Rendering

The prior works from DTA and ACTIVE[18; 19] prove that realistic rendering of the vehicle camouflage is one of the keys to successful physical adversarial attack. To achieve this, we use two rendering components: the first render component is a differentiable neural renderer, which takes the 3D properties of the vehicle to output the vehicle's foreground images. However, it struggles to render complex environmental characteristics on the vehicle surface. To alleviate this, we use an environmental feature renderer that can combine the environmental characteristics and neural renderer output to produce realistic and accurate camouflaged vehicle images.

We select Pytorch3D as differential renderer [12] to generate camouflaged vehicle images because it supports differentiable path to the UV-map texture image. Following the method proposed in DTA [18] and ACTIVE [19], we use a U-Net network for EFR to extract environmental characteristics from $X_{ref}$ and combine them with $X_{nr}$. EFR outputs the camouflaged vehicle with environmental characteristics $X_{ren}$.

Before camouflage generation, we need to train EFR for its optimal performance. The training of EFR needs masked vehicle images $X_{ref}$, 3D mesh $Msh$, camera positions $\Phi_{cam}$, and various preset color texture $T$ as input. Meanwhile, we obtain images of different preset colors from the Carla simulation environment. Then, we mask out the vehicle parts as the network ground truth $GT$. To optimize the parameters of EFR, we use the following loss function:

$$L_{EFR}(X_{ref}) = W(X_{ref})\, BCE(X_{ren}, GT), \tag{6}$$

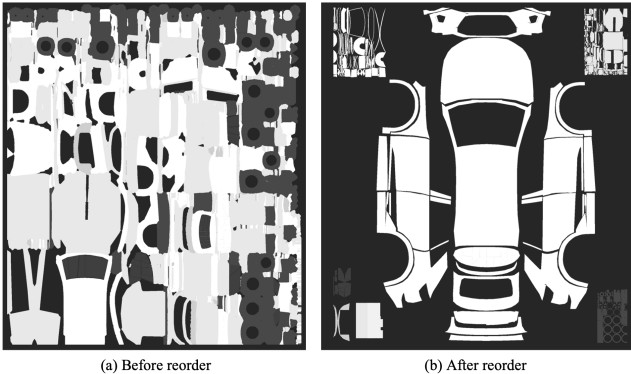

(a) Before reorder          (b) After reorder

Figure 3: Reordered texture UV map to improve camouflage naturalness.

where BCE is the binary cross-entropy loss, and $W(X_{ref}) = \frac{H \cdot W}{S}$ is a weight function. $H$ and $W$ are the heights and widths of $X_{ref}$, and $S$ is the number of pixel points in the vehicle part of $X_{ref}$. $W(X_{ref})$ can balance EFR rendering optimization across various camera angles, especially for views where the vehicle occupies a small area of the image.

### 3.3 Adversarial Texture Generation with Diffusion Model

Prior works optimize the adversarial camouflage in the pixel space, which leads to unnatural and uncontrollable adversarial texture patterns. To alleviate this, we leverage an off-the-shelf stable diffusion model [15] to generate a vehicle UV-map texture image. The diffusion model is trained on a subset of LAION-5B [16], a dataset of billions of image-text pairs. Since the diffusion model learns the manifold of natural images and corresponding text captions, it can generate adversarial texture images that look more natural and relevant to the given text prompt.

To make the generated UV-map images adversarial, we introduce an adversarial feature vector $F_{adv}$ that can be optimized during the camouflage generation with diffusion models. The adversarial feature vector has the same hidden dimension as the text feature vector $F_{txt}$. Therefore, it can concatenate with $F_{txt}$ to form the conditional input to the T2I model. As a result, the generated image reflects the control signal from the text feature while being adversarial against the target detector.

We also reordered the UV map of the vehicle texture so that the vehicle surface could be connected as much as possible. Figure 3 shows the UV mapping before and after reordering. The reordering makes the generated camouflage can keep more natural patterns generated by the diffusion model. Hence, it improves the naturalness of the vehicle camouflage.

### 3.4 Naturalness vs Attack Performance Trade-off

Without any constraints, the framework can move the combined feature from $F_{txt}$ and $F_{adv}$ out of the feature space of the text prompt. Consequently, we can no longer expect the generated UV-map images to look natural and consistent with the input text prompt. Since the diffusion model is trained to generate natural and controllable images with original text features, there is a higher chance of generating natural images if the concatenated feature is closer to the original text feature.

To preserve naturalness and controllability, we assure that the adversarial feature $F_{adv}$ will not exceed a norm greater than a threshold $\tau$. Tuning the norm threshold $\tau$ enables us to trade off naturalness and controllability for attack performance.

We follow PGD and choose $\ell_p$ norm to constrain $F_{adv}$. We update $F_{adv}$ using the formula below:

$$F_{adv}^t = \kappa \left( F_{adv}^{t-1} + \eta \nabla L_{adv} \right), \tag{7}$$

$$\kappa(F) = \{ F_i \mid F_i \leftarrow \min \left( \max \left( F_i, -\tau \right), \tau \right), F_i \sim F \} \tag{8}$$

where $t$ is the time step, $\eta$ is the step size, $\nabla L_{adv}$ is the gradient of the adversarial loss, $\kappa$ is the clipping function defined as in Eq. 8, where $F_i$ is the $i$-th element of $F$.

# 4 Experiments

## 4.1 Experimental Settings

**Datasets**: We utilize the Carla [3] simulator to generate datasets for our experiments. To have a comparative analysis with prior studies [22; 21; 18; 19], we select the Audi E-Tron as the target vehicle model. We create datasets using various simulation settings, resulting in 69,120 and 59,152 photo-realistic images for EFR training and testing. These images cover 16 distinct weather conditions, combining four sun altitudes and four fog densities. Additionally, we generate a dataset of 40,960 images for camouflage generation and a test dataset of 8192 images for adversarial camouflage evaluation. The weather conditions included in these two datasets are the same as those used during the training and testing of EFR. We print out five types of adversarial camouflages for physical world evaluation and apply them to 1:12 Audi E-Tron car models. For each model, we capture 96 pictures under different elevations, azimuths, and distance parameters.

**Baselines**: We compare our method with four advanced adversarial camouflage methods: DAS [22], FCA [21], DTA [18], and ACTIVE [19]. DAS and FCA optimize the 3D texture by minimizing the attention-map scores and detection scores of the detector, respectively. DTA and ACTIVE both optimize a square texture pattern with a neural network and cover it on the vehicle's surface repeatedly. We compare our results using the official textures generated by these methods. We apply these textures to the same car model for a fair comparsion.

**Evaluation metrics**: For training the EFR component, we follow the setting from [18] to use the Mean Absolute Error (MAE) as a loss to measure the difference between the output of ERP and the ground truth. To evaluate the effectiveness of the adversarial camouflage, we utilize the AP@0.5 benchmark [4], as it provides a comprehensive assessment of recall and precision values at a detection IOU threshold of 0.5.

**Target detection models**: Aligning with previous work, we adopt YOLOv3 [13] as the white-box target detection model for adversarial camouflage generation. To evaluate the effectiveness of the optimized camouflage, we utilize a collection of widely used object detection models treated as black-box models, including YOLOF [2], Deformable DETR (DDTR) [27], Dynamic R-CNN (DRCN) [24], Sparse R-CNN (SRCN) [17], and Faster R-CNN (FrRCNN) [14]. They are trained on the COCO dataset and implemented in MMDetection [1].

**Training details**: Following [18], we utilize the Adam optimizer with a learning rate 0.01 for EFR training and camouflage generation. We train the EFR for 20 epochs and choose the model with the best performance on the test dataset. We use stable diffusion v1.5 to generate UV-map texture image with DDIM sampler. We set the sampling step to 20. The optimization of the adversarial camouflage takes a duration of five epochs. We conduct experiments on a cluster with eight NVIDIA RTX A800 80GB GPUs.

## 4.2 Attack Performance Evaluation

### 4.2.1 Attack in the Digital World

In this section, we compare our method to current advanced adversarial camouflage methods, including DAS [22], FCA [21], DTA [18], and ACTIVE [19]. We run an extensive attack comparison using diverse detection models. Since the target model is YOLOv3, we use various detection models to evaluate the transferability of the camouflage in a black-box setting. We use *'colorful camouflage'* as the text prompt in this experiment.

The results are shown in Table 1, showing that our method has competitive performance with the current state-of-the-art baselines. DAS performs only better than normal car painting, primarily due to the limitations of partially painted camouflage. Meanwhile, FCA exhibits sub-optimal performance, only slightly better than random camouflage, because it cannot render sophisticated environment characteristics. DTA and ACTIVE have comparable attack performance to CNCA, but our method achieves the best attack performance in total. Figure 4 shows the summarized performance of each camera pose and weather parameter; values are car AP@0.5 averaged from the detectors used in Table 1. We can see that the camouflage produced by our method shows competitive performance compared to DTA and ACTIVE.

Table 1: Comparison of the effectiveness of camouflages across various object detection models. Values are AP@0.5 of the car.

| METHODS | SINGLE-STAGE | | | TWO-STAGE | | | TOTAL |
|---|---|---|---|---|---|---|---|
| | YOLOv3 | YOLOF | DDTR | DRCN | SRCN | FRRCN | |
| NORMAL | 0.712 | 0.824 | 0.803 | 0.778 | 0.786 | 0.771 | 0.779 |
| RANDOM | 0.642 | 0.753 | 0.625 | 0.694 | 0.681 | 0.672 | 0.678 |
| DAS | 0.671 | 0.769 | 0.738 | 0.715 | 0.724 | 0.719 | 0.723 |
| FCA | 0.581 | 0.725 | 0.603 | 0.678 | 0.642 | 0.668 | 0.650 |
| DTA | 0.521 | 0.657 | **0.402** | 0.614 | 0.488 | 0.562 | 0.541 |
| ACTIVE | **0.473** | 0.577 | 0.436 | **0.534** | 0.484 | 0.520 | 0.504 |
| CNCA | 0.485 | **0.538** | 0.436 | 0.536 | **0.470** | **0.504** | **0.495** |

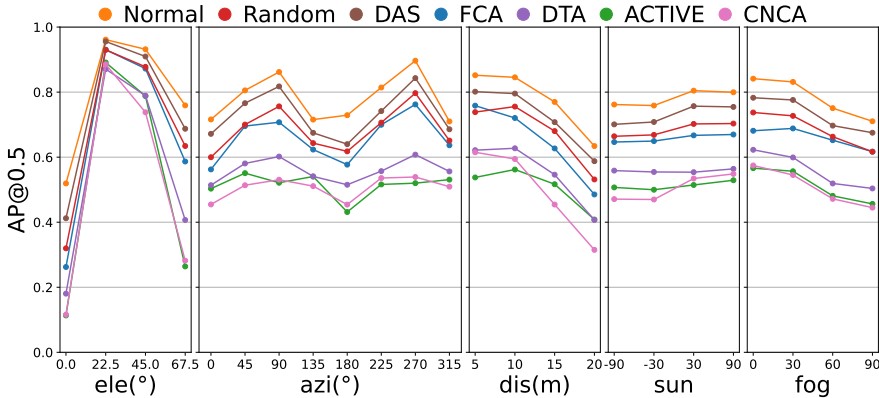

Figure 4: Attack comparison on different camera poses and weather parameters. "ele" denotes elevation, "azi" denotes azimuth, "dis" denotes distance, "fog" denotes fog density, and "sun" denotes sun altitude angle. Values are car AP@0.5 (%) averaged from all models.

### 4.2.2 Attack in the Physical World

We conduct the physical world evaluation using 1:12 Audi E-Tron car models with camouflages generated by different methods. Table 2 shows the attack performance of these camouflages against real-time object detectors in the real world: YOLOv3, YOLOX [5], SSD [10], CenterNet [26], RetinaNet [9]. Our method achieves the best attack performance against four out of five detectors and the best overall performance. Furthermore, Figure 5 shows that our method can successfully attack the detectors in both indoor and outdoor environments compared to other methods. In summary, the physical evaluation results demonstrate that our method is transferable to the real world.

### 4.3 Customzablility and Naturalness Evaluation

### 4.3.1 Customizable Camouflage Generation

Our method enables customizable camouflage generation with user-specific input. Table 3 shows the various styles of adversarial camouflages with their corresponding text prompts and AP@05 values (target YOLOv3). Our method can directly customize the color choices and patterns of the camouflage with the input texture prompt. We notice that there are some differences in attack performance regarding different prompts. During our experiments, we found the prompts that describe natural objects (columns 4,5,6 in Table 3) are likely to have lower attack performance(average around 0.03 in AP@0.5) than the more abstract ones (columns 1,2,3 in Table 3). Despite this, the average AP@0.5 of all these camouflages generated by CNCA is 0.506, comparable to the reported result in Table 1.

Table 2: AP@0.5 of the different methods in the physical world evaluation.

| METHODS | YOLOv3 | YOLOX | SSD | CENTERNET | RETINANET | TOTAL |
|---|---|---|---|---|---|---|
| NORMAL | 0.778 | 0.936 | 0.890 | 0.916 | 0.981 | 0.900 |
| DAS | 0.734 | 0.878 | 0.847 | 0.873 | 0.955 | 0.857 |
| FCA | 0.560 | 0.770 | 0.767 | 0.798 | 0.921 | 0.763 |
| DTA | 0.566 | 0.689 | 0.786 | 0.854 | 0.886 | 0.756 |
| ACTIVE | 0.518 | 0.563 | 0.574 | 0.743 | **0.735** | 0.627 |
| CNCA | **0.439** | **0.464** | **0.557** | **0.698** | 0.780 | **0.588** |

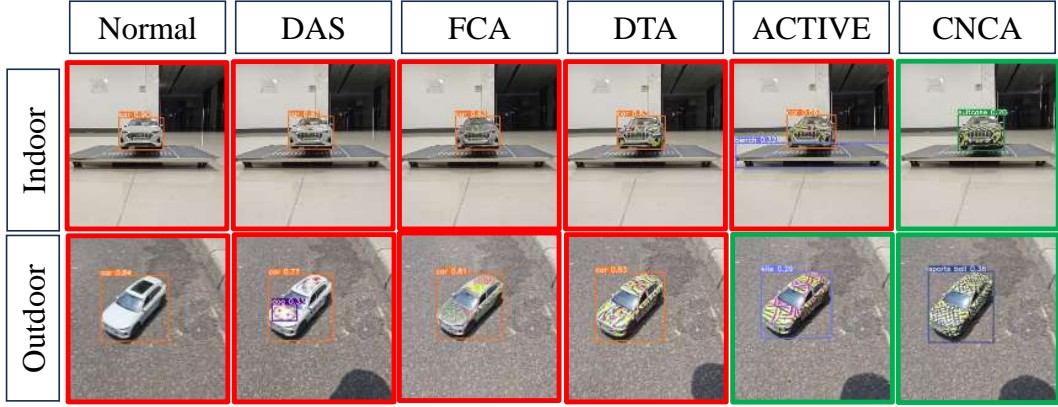

Figure 5: Examples of real-world evaluation for different methods. The evaluation includes both indoor and outdoor environments.

Table 3: Customizable camouflages with different text prompts. The AP@0.5 of each camouflage is shown below. The normal car texture baseline is 0.712.

| CUSTOMIZABLE CAMOUFLAGE GENERATION WITH USER-SPECIFIC TEXT PROMPT. | | | | | |
|---|---|---|---|---|---|
| *colorful graffiti* | *yellow black graffiti* | *colorful camouflage* | *colorful balls* | *snake texture* | *zebra strips* |
| 0.508 | 0.479 | 0.485 | 0.516 | 0.486 | 0.561 |

Table 4: Subjective tests for the naturalness evaluation of our adversarial camouflage with other baselines. The naturalness score is scaled from 1(not natural at all) to 5(very natural). Besides reporting mean and standard deviations of each type of camouflage, we also conduct t-tests and report the t and p values to verify the differences in means scores are significant (p value less than 0.05). As shown in the results, our method's score is significantly higher than the previous four methods.

| IMAGES | | | | | | |
|---|---|---|---|---|---|---|
| SCORE | $4.68 \pm 0.67$ | $2.05 \pm 1.03$ | $2.11 \pm 1.17$ | $1.86 \pm 0.98$ | $1.84 \pm 0.99$ | $2.84 \pm 1.09$ |
| T VALUE | 8.72 | -3.18 | -2.77 | -4.04 | -4.13 | NA |
| P VALUE | < 0.00001 | 0.00108 | 0.00358 | 0.00007 | 0.00005 | NA |
| SOURCE | NORMAL | DAS | FCA | DTA | ACTIVE | CNCA(OURS) |

Table 5: Ablation study of CNCA pipeline components. The input text prompt is "yellow, black graffiti." We evaluate each test pipeline in terms of attack performance and naturalness. The description of each test pipeline is the following: No Diff: we directly optimize the UV map texture of the vehicle without the integration of the diffusion model; Diff.: we use the diffusion model to generate UV-map texture but without introducing adversarial feature; Diff.+ Adv. Introduces the adversarial feature but without the clipping strategy; Diff.+Adv.+Clip introduces clipping strategy but without reordering of texture map; Diff.Adv.+Clip+Reorder is our final pipeline with reordering the texture mask.

| PIPELINE | NO DIFF. | DIFF. | DIFF.+ ADV. | DIFF.+ ADV. + CLIP | DIFF. + ADV. + CLIP + REORDER |
|---|---|---|---|---|---|
| TEXTURE |  |  |  |  |  |
| AP@0.5 | 0.619 | 0.553 | 0.520 | 0.494 | 0.479 |
| SCORE | 1 | 3.37 | 1.71 | 2.75 | 3.33 |

#### 4.3.2 Naturalness Score by Subjective Evaluation

Our proposed approach aims to improve the naturalness of the generated adversarial camouflage to humans. Therefore, following [7; 8], we conducted a subjective evaluation to estimate the naturalness score of the adversarial camouflages. For a fair comparison, we generate a series of vehicle images using the same set of camera positions for each type of camouflage. Then, we require each participant to give a naturalness score for each type of camouflage from a scale of 1 (not natural at all) to 5 (very natural). Besides the advanced adversarial camouflages from prior work, we also include the normal car texture as the control group. To further demonstrate the significance of the differences in mean scores, we also conduct t-tests and report the t and p values. As shown in Table 4, the p values of t-tests are lower than 0.05. Therefore, our method's naturalness score is significantly higher than those of the four adversarial camouflages.

### 4.4 Ablation Studies

**The impact of each CNCA pipeline component.** Table 5 shows the results of the ablation studies for each component of the pipeline. We gradually add each component during the ablation study to see their contribution to attack performance and naturalness. All the test pipelines with the diffusion model use the same input text prompt: "yellow black graffiti." The discussion for each test pipeline is the following: No Diff. directly optimizes the texture image of the vehicle at a pixel level, resulting in an unnatural texture; Diff. introduces the diffusion model to generate the texture image compared to No Diff., which improves the naturalness score; Diff.+Adv. introduces the adversarial feature compared to Diff., which enables the texture image generation guided by the adversarial gradient from the detector. The adversarial feature improves the attack performance but compromises the naturalness; Diff.+Adv.+Clip introduces the clipping strategy compared to Diff.+Adv., which improves the naturalness; Diff.+Adv.+Clip+Reorder is our final pipeline, which uses a reordered texture map compared to Diff.+Adv.+Clip, which further improves the attack performance and naturalness. To summarize, the ablation studies demonstrate that all the components of our pipeline contribute to improving the camouflage's attack performance and naturalness.

**Trade-off between Naturalness and Attack Performance.** There is inevitably a trade-off between naturalness and attack performance. The optimization space for adversarial attacks decreases when increasing naturalness of the attack. Therefore, increasing naturalness will typically decrease attack performance. Our method allows users to balance this trade-off based on their preference by adjusting the norm threshold $\tau$. To illustrate this trade-off, we generated adversarial UV texture images with the same text prompt but different norm threshold settings for the adversarial features. In addition, we conduct a subjective survey to rank the naturalness of the images and their relevance regarding the text prompt. Table 6 shows the average rank of each image, its corresponding AP@0.5, and the norm threshold settings. It can be seen that when the norm threshold starts to increase, the texture images become less natural-looking, and the attack performance increases. Therefore, it requires the user to decide the acceptable naturalness level.

Table 6: Naturalness test average score against attack performance using YOLOv3. The input text prompt is "yellow black graffiti".

| Threshold | 0.1 | 0.5 | 1 | 1.5 | ∞ |
|---|---|---|---|---|---|
| Texture |  |  |  |  |  |
| AP@0.5 | 0.606 | 0.449 | 0.479 | 0.481 | 0.520 |
| Score | 4.79 | 3.54 | 3.33 | 1.46 | 1.71 |

## 5  Limitations & Societal Impact

**Limitations.** Our current method has the following limitations: firstly, the back-propagation is more expensive than the previous gradient-based methods, but we believe the strong prior knowledge of the diffusion model can further advance adversarial attacks and defenses; secondly, our method needs the 3D mesh of the target object to generate camouflage. If the 3D mesh of the object is not available, the user needs additional efforts such as photogrammetry or 3D scanning to obtain it; lastly, for each text prompt, a manual hyperparameter tuning for norm threshold $\tau$ is needed. One future work is to automatically adjust $\tau$ based on the relevance score between the image and text prompt, for instance, CLIP score [11].

**Societal Impact.** This paper presents work whose goal is to advance the safety of AI systems. While the proposed adversarial attack method could be potentially used by malicious users, it can also support future efforts to enhance the robustness of AI system via adversarial training, adversarial testing and adversarial example detection, thereby safeguarding the security of AI systems.

## 6  Conclusion

We propose a novel physical adversarial camouflage attack framework with diffusion models. With different user-specific text prompts, our method can generate adversarial camouflage with diverse colors and patterns. In particular, we apply a clipping strategy to an adversarial feature to balance the adversarial camouflage's naturalness and attack performance. With extensive experiments on the digital and physical world and user studies, the results demonstrate that our methods improve the naturalness and enable the customizability of camouflage generation while maintaining competitive attack performance.

## Acknowledgment

This research is supported by the National Natural Science Foundation of China (Grant: 62306093, 62376074), the Guangdong Provincial Key Laboratory of Novel Security Intelligence Technologies (Grant No. 2022B1212010005), the Shenzhen Science and Technology Program (Grants:JSGGKQTD20221101115655027, RKX20231110090859012, SGDX20230116091244004), the Fundamental Research Funds for the Central Universities (Grant No. HIT.OCEF.2024047), and the Fok Ying Tung Education Foundation of China (Grant 171058). Daojing He and Yu Li are the corresponding author of this article.

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

# A   Subjective Human Evaluation

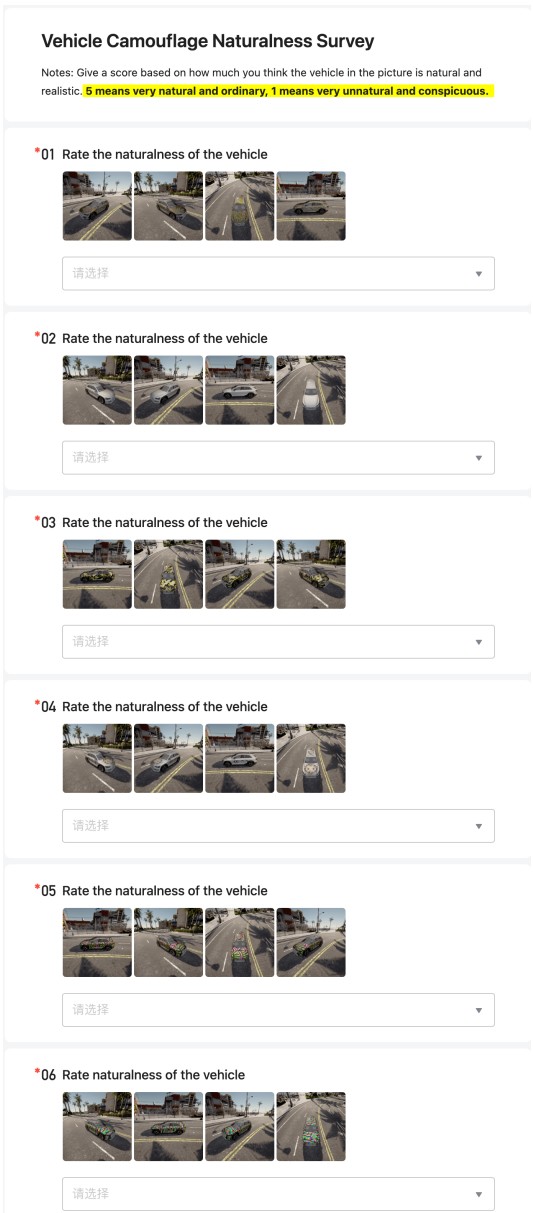

Figure 6: The interface of our Survey for Human-Evaluations: We present the participants with several pictures of the camouflaged vehicle for each method. The order of the methods is random. We ask the participants to rate the naturalness of the camouflage on a scale of 1 to 5.

To better evaluate the naturalness of CNCA compared with another adversarial camouflage attack, we conducted a survey among humans with the assistance of an online form. The user interface of the survey is shown in Figure 6, where the participants are asked to give a 1(very unnatural) to 5(natural) rating regarding the naturalness of the vehicle's appearance. We collect surveys from 45 participants up to the completion of the writing and most of them are not familiar with adversarial attacks. Table 4 illustrates the results, indicating that CNCA achieves higher naturalness among human participants compared with previous advanced adversarial camouflage.

