# OpenReview forum: "CNCA: Toward Customizable and Natural Generation of Adversarial Camouflage for Vehicle Detectors"
_NeurIPS.cc/2024/Conference — NeurIPS 2024 poster_

### Official Review · Reviewer_oknY · 2024-07-02

**Soundness:** 2
**Presentation:** 3
**Contribution:** 3
**Rating:** 5
**Confidence:** 5

**Summary:**

This paper presents a novel method CNCA for generating customizable and natural adversarial camouflage of fooling vehicle detectors. This work is an interesting contribution in the field of adversarial attacks, especially improving the naturalness of the camouflage while maintaining high attack performance.

**Strengths:**

It is interesting to apply diffusion models to physical adversarial attacks to generate natural camouflage for the first time, and it is also of practical value to generate natural adversarial camouflage with customizable styles based on text prompts.

**Weaknesses:**

The clipping strategy lacks innovation; it has been used in PGD for a long time, and it is not worth spending too much space on it.

**Questions:**

1. Why is there no comparison with baselines in the physical world?
2. Whether other detection models can be used as white-box to generate adversarial textures to verify the transferability of CNCA.
3. Only use the subjective evaluation may not be convincing for experimental verification, and visual observation does not seem to be more natural than previous methods. Is there a more convincing scoring rule to evaluate naturalness?

**Limitations:**

The experimental results are insignificant.

---

> ### Author Rebuttal · Authors · 2024-08-06
>
> **W1: The clipping strategy lacks innovation; it has been used in PGD for a long time, and it is not worth spending too much space on it.**
>
> The core contribution of our work is introducing the diffusion model to enable the customizable and natural generation of physical adversarial camouflage. The clipping strategy from PGD itself is not novel, but our contribution is adapting it to regulate the perturbation level of the diffusion model. We describe the details of clipping ( in only 3 lines) because we want to make our method clear to readers who are not familiar it.
>
> **Q1: Why is there no comparison with baselines in the physical world?**
>
> For our physical world evaluation, we follow the setting from the previous state-of-the-art works of DTA and ACTIVE, which only compares the performance between the normal and the generated camouflage. For the completeness of the evaluation, we extend the physical evaluation with previous baselines and both indoor and outdoor environments. The results are as shown in the table below:
>
> | **Methods** | YOLOv3 | YOLOX | SSD  | CenterNet | RetinaNet | Total |
> |-------------|--------|-------|------|-----------|-----------|-------|
> | **Normal**  | 0.778  | 0.936 | 0.890| 0.916     | 0.981     | 0.900 |
> | **DAS**     | 0.734  | 0.878 | 0.847| 0.873     | 0.955     | 0.857 |
> | **FCA**     | 0.560  | 0.770 | 0.767| 0.798     | 0.921     | 0.763 |
> | **DTA**     | 0.566  | 0.689 | 0.786| 0.854     | 0.886     | 0.756 |
> | **ACTIVE**  | 0.518  | 0.563 | 0.574| 0.743     | **0.735** | 0.627 |
> | **CNCA**    | **0.439** | **0.464** | **0.557** | **0.698** | 0.780 | **0.588** |
>
>
> The results demonstrate that our method achieves comparable performance with previous baselines in the real world, which matches the results from the digital world evaluation. The details of the extended physical evaluation can be found in the G2 section of the global rebuttal. The physical examples can be found in Figure 1 in the attached PDF.
>
> **Q2: Whether other detection models can be used as white-box to generate adversarial textures to verify the transferability of CNCA.**
>
> Yes, they can. We currently use YOLOv3 to generate adversarial camouflage for a fair comparison with previous methods (FCA, DTA, and ACTIVE), which also use this detector to generate camouflage. To validate the transferability, we use YOLOv5 as the attacked white-box detector to generate camouflage. We keep the other experiment settings the same, such as input prompt (yellow black graffiti) and clipping threshold (value is 1). We report the attack and naturalness performance using the average of car AP@0.5 over 5 different detectors and human evaluation. The results are as follows:
>
> |  Attack Detector             | YOLOv3 | YOLOv5 |
> |----------------|--------|--------|
> | Averaged AP@0.5         | 0.522  | 0.518  |
> | Natural Score  | 3.33   | 4.00      |
>
> The results demonstrate that CNCA has similar attack performance regardless of the white-box models in the CNCA framework, which verifies its transferability.
>
> **Q3: Subjective evaluation may not be convincing for experimental verification, and visual observation does not seem to be more natural than previous methods. Is there a more convincing scoring rule to evaluate naturalness?**
>
> Evaluating the naturalness of physical adversarial attacks is a challenging task. We have surveyed the existing work related to this task. Among these works, S. Li et al. [1] is the most relevant one because they evaluate the naturalness of the physical attack for vehicle detection. They trained an evalution model with vehicle images and the corresponding human ratings. However, we found the trained model shows a bias of low scoring towards the full-covered painting vehicle image, even if the vehicle painting is designed by humans and looks very natural. As a result, all the full-cover baselines receive low scores. We believe it is caused by the training dataset's lack of full-cover painting vehicle images. To avoid this bias, we decide to follow the work of [2] and conduct a subjective survey to directly evaluate the naturalness by considering full-cover painting vehicles (for instance, racing cars with banners and logos) still natural. To maintain the fairness of the survey, we invited 45 participants from different ages, backgrounds, and genders, which is twice the number in [3] ( which is 24).
>
> [1] S. Li et al., “Towards Benchmarking and Assessing Visual Naturalness of Physical World Adversarial Attacks,” In Proceedings of the IEEE/CVF Conference on Computer Vision and Pattern Recognition. 2023, pp. 12324–12333, May 2023.
>
> [2] Hu, Yu-Chih-Tuan, et al. "Naturalistic physical adversarial patch for object detectors." Proceedings of the IEEE/CVF International Conference on Computer Vision. 2021.
>
> **L1: The experimental results are insignificant.**
>
> Our primary goal is to improve naturalness and enable customization of the physical adversarial camouflage. It is challenging to achieve this goal while maintaining competitive attack performance. We leverage the pre-trained diffusion model to generate adversarial texture images with input prompts to tackle this. Besides, we introduce adversarial features so that the adversarial gradient from the detector can guide the camouflage generation. Finally, we adapt the clipping strategy to balance the trade-off between attack performance and naturalness. **Experimental results show that our attack performance is close to the state-of-the-art method (ACTIVE), but our method's naturalness score is 54% higher, which is a significant improvement**. While our work isn't perfect, it represents a significant initial effort towards customizable and natural camouflage generation. We believe it will draw increased research attention to this field.

---

> ### Author Response · Authors · 2024-08-14
> **Reply to Reviewer oknY**
>
> Dear Reviewer oknY,
>
> Thank you for your time and effort in reviewing our work and rebuttal. Your feedback has helped us improve. We are grateful that you raised the rating for our work after we provided the clarifications in the rebuttal!
>
> Best Regards，
>
> Authors of Submission 17110

---

### Official Review · Reviewer_MuNz · 2024-07-06

**Soundness:** 3
**Presentation:** 2
**Contribution:** 3
**Rating:** 5
**Confidence:** 2

**Summary:**

The paper introduces a interesting idea and also a novel method called Customizable and Natural Camouflage Attack (CNCA) to generate adversarial camouflage against vehicle detectors, leveraging a pre-trained diffusion model. This approach allows the generation of natural-looking and user-customizable adversarial patterns that maintain robust attack performance across various digital and physical settings. The paper's contributions include a unique application of diffusion models to adversarial camouflage, introduction of adversarial features for gradient-based generation, and a clipping strategy to balance naturalness with attack performance. Extensive experiments and user studies demonstrate the effectiveness of CNCA in producing more natural-looking camouflage with competitive attack performance.

**Strengths:**

- The paper proposes an interesting and useful application direction, namely natural and customized adversarial camouflage. The research motivation has substantial practical significance, and the proposed method appears intuitively reasonable.
- This study is the first to apply diffusion models for natural adversarial camouflage generation. It is also the first to generate various 52 styles of adversarial camouflage against vehicle detectors.
- The experiments are thorough, and the results are statistically significant, indicating high-quality research.
- The code is provided.

**Weaknesses:**

- Some expressions are not clear, making it difficult for those unfamiliar with the field to understand. For example, lines 32 to 35. It would be better and easier to understand if some visual evidence were provided regarding these limits.
- Figure 1 is not correctly referenced.
- The experimental section would be more convincing if the effectiveness of the proposed components and methods were demonstrated through ablation experiments.
- Concerning anonymity: Some comments in the provided code reveal personal information. Please be aware of this!

**Questions:**

- I am curious about its complexity. The method involves multiple components and parameters (e.g., adversarial features, clipping strategy), which might complicate its deployment in practical applications without substantial customization and tuning.
- It would be more helpful if more ablation experiments were added to individually demonstrate the functions of each component.

**Limitations:**

Refer to the “Questions” section.

---

> ### Author Rebuttal · Authors · 2024-08-06
>
> **W1: lines 32-35 are not clear to understand. Provide visual evidence to illustrate.**
>
> We would like to clarify that lines 32-35 explain the two reasons why the previous camouflage methods lack naturalness. Firstly, these methods lack prior knowledge of naturalness to guide the camouflage generation. Secondly, these methods optimize the adversarial texture at a pixel level, making it difficult to form a natural-looking texture pattern. For instance, the previous state-of-the-art methods like FCA, DTA, and ACTIVE generate suspicious and attention-grabbing patterns, resulting in a low-score performance in naturalness, as shown in Table 4 in our paper.
>
> **W2: Figure 1 is not correctly referenced.**
>
> We will correct it in future versions of our paper.
>
>
> **W3: The experimental section would be more convincing if the effectiveness of the proposed components and methods were demonstrated through ablation experiments.**
>
> | Pipeline                    | No Diff. | Diff. | Diff. + Adv. | Diff. + Adv. + Clip | Diff. + Adv. + Clip + Reorder |
> |-----------------------------|----------|-------|--------------|---------------------|------------------------------|
> | AP@0.5                 | 0.619    | 0.553 | 0.520        | 0.494               | 0.479                        |
> | Natural Score                   | 1.00        | 3.37  | 1.71         | 2.75                | 3.33                         |
>
>
> The above table shows the results of the ablation studies for each component of the pipeline. During the ablation study, we gradually add the components individually to see their contribution to attack performance and naturalness. All the test pipelines with the diffusion model use the same input text prompt: "yellow black graffiti." The description for each test pipeline is the following:
>
> - **No Diff.** directly optimizes the texture image of the vehicle at a pixel level, resulting in an unnatural texture;
>
> - **Diff.** introduces the diffusion model to generate texture image compared to **No Diff.**, which improves the naturalness score;
>
> - **Diff.+Adv.** introduces the adversarial feature compared to **Diff.**. The attack performance improves but compromises the naturalness;
>
> - **Diff.+Adv.+Clip** introduces the clipping strategy compared. to **Diff.+Adv.**, which improves the naturalness;
>
> - **Diff.+Adv.+Clip+Reorder** is our final pipeline, which uses a reordered texture map compared to **Diff.+Adv.+Clip**, which further improves the attack performance and naturalness.
>
> **No Diff.** optimizes the texture image of the vehicle with no prior knowledge of naturalness at a pixel level, which results in an unnatural texture image. To improve naturalness, **Diff.** leverages the prior knowledge from the diffusion model. **Diff.+Adv.** introduces the adversarial feature, which enables the texture image generation guided by the adversarial gradient from the detector. It improves the attack performance but compromises the naturalness. **Diff.+Adv.+Clip** introduces the clipping strategy to regulate the perturbation level of diffusion, which improves the naturalness. Our final pipeline **Diff.+Adv.+Clip+Reorder** uses a reorder texture map to keep more natural patterns generated by the diffusion model, further improving the attack performance and naturalness.
>
> To summarize, the ablation studies demonstrate that all the components of our pipeline contribute to improving the camouflage's attack performance and naturalness. Please find the above table and corresponding generated texture images in Table 1 from the attached PDF.
>
>
> **W4: Concerning anonymity: Some comments in the provided code reveal personal information.**
>
> We have made the necessary modifications to remove the personal information in the code.
>
> **Q1: The complexity during its deployment in practical applications without substantial customization and tuning.**
>
> In the early stage of our experiment, we indeed spent some effort manually tuning the clipping threshold to generate the adversarial camouflage for a certain prompt. To amend this, we propose an automatic tunning method to adjust the clipping threshold dynamically. We calculate the relevance score between the input prompt and the generated image. If the relevance score is high above the pre-defined threshold, the clip threshold will use a larger value, allowing more exploration of the adversarial feature. If the relevance score is lower than the threshold, which means the generated images cannot match the input prompt, the clipping threshold will use a smaller value to constrain the adversarial feature. To calculate the relevance score, we leverage the CLIP[1] model,  an effective pre-trained model that learns the relevance between visual images and text captions.
>
> To validate this idea, we conducted an ablation study on manual and automatic tunning under the same input prompt ("yellow black graffiti") and detection model (YOLOv3) settings, showing that they achieve comparable attack and naturalness performance measured by AP@0.5 and naturalness score, as shown in the table below.
>
> |     Tuning Method               | Manual Tuning | Automatic Tuning |
> |--------------------|---------------|------------------|
> | AP@0.5| 0.479         | 0.509            |
> | Natural Score        | 3.33          | 3.28             |
>
> [1] Radford, Alec, et al. "Learning transferable visual models from natural language supervision." International conference on machine learning. PMLR, 2021.
>
> **Q2: Add more ablation experiments to demonstrate the functions of each component individually.**
>
> Please refer to the reply to W3.

---

> ### Author Response · Authors · 2024-08-14
> **Friendly Reminder: Follow-Up on Rebuttal for Submission 17110**
>
> Dear Reviewer MuNz,
>
> We are writing to follow up on the rebuttal we submitted regarding your review comments for our paper. We appreciate your time and effort in reviewing our work and providing valuable feedback. We have made a sincere effort to address each of your comments and questions in the rebuttal. We believe the clarifications and improvements we made in response to your suggestions have strengthened the paper significantly.
>
> We kindly request that you review our rebuttal as soon as possible ( today is the final day for discussion ) and consider increasing your rating for our paper with the provided changes and clarifications. Thank you again for your dedication to the review process; we look forward to hearing from you!
>
> Best Regards，
>
> Authors of Submission 17110

---

> > ### Comment · Reviewer_MuNz · 2024-08-14
> >
> > The author addressed most of my concerns, I will rise the score.

---

### Official Review · Reviewer_guSy · 2024-07-11

**Soundness:** 3
**Presentation:** 3
**Contribution:** 2
**Rating:** 4
**Confidence:** 5

**Summary:**

The manuscript presents a novel approach to generating physical adversarial camouflage against vehicle detectors, leveraging a pre-trained diffusion model. The proposed method, called Customizable and Natural Camouflage Attack (CNCA), aims to produce adversarial camouflage that is both natural-looking and customizable via user-specific text prompts. This approach addresses the limitations of previous methods that produced conspicuous and unnatural camouflage, maintaining effectiveness in adversarial attacks while enhancing the camouflage's appearance to blend seamlessly into its surroundings.

**Strengths:**

CNCA introduces a novel application of diffusion models for generating physical adversarial camouflage, a significant shift from the traditional pixel-level optimization methods.

The method allows for the generation of camouflage that is not only effective in evading detection but also customizable and more natural-looking, meeting specific user requirements.

The manuscript provides a comprehensive evaluation of the CNCA approach, including both digital and physical world tests and user studies, demonstrating its effectiveness and practical applicability.

**Weaknesses:**

The approach involves complex integration of diffusion models with adversarial attack frameworks, which may increase the computational overhead and complexity compared to more straightforward adversarial techniques.

Although the manuscript includes extensive testing, the evaluations focus primarily on vehicle detection in controlled settings. The performance and practicality of CNCA in more varied or less controlled environments remain to be fully explored.

**Questions:**

See strength and weakness above.

**Limitations:**

No. While the paper discusses potential positive impacts, such as improving AI robustness, the technique could also be used maliciously to evade surveillance, posing ethical and security concerns.

---

> ### Author Rebuttal · Authors · 2024-08-05
>
> **W1: Integration of Diffusion models with adversarial attack frameworks may increase computational overhead and complexity.**
>
> We have discussed this weakness in the section on Limitations & Societal Impact. We would like to clarify that our work's novelty enables the naturalness and customizability of the physical adversarial attack. Integration with diffusion models is the key to achieving this. Despite its higher computational cost, camouflage can be generated offline at a one-time cost. The generated camouflage can be used to attack a wide range of detectors. In summary, introducing diffusion models is still valuable, although it increases the computation cost.
>
> **W2: The evaluations focus primarily on vehicle detection in controlled settings. The performance and practicality of CNCA remain to be fully explored in more varied or less controlled environments .**
>
> Vehicle detection is crucial in autonomous driving and traffic monitoring. Therefore, many previous methods chose to research the physical attack on this task setting. Our work follows this research direction. We agree that we need to validate our method's performance and practicability in more varied environments. Therefore, we extend our physical evaluation to both indoor and outdoor environments, as shown in Figure 1 in the attached PDF. The results demonstrate that our method is transferable in varied real-world environments.
>
> **L1: The technique could also be used maliciously to evade surveillance, posing ethical and security concerns.**
>
> We have discussed this in the section of Limitations & Societal Impact. We acknowledge the potential malicious usage of our technique. However, although our method generates physical attack examples, these examples can be used for the research of defense methods, such as adversarial training, adversarial testing, and adversarial example detection. The research of defense can ultimately safeguard AI systems.

---

> > ### Comment · Reviewer_guSy · 2024-08-12
> >
> > The author's response addressed some of my questions and I decided to keep my rating.

---

> ### Author Response · Authors · 2024-08-14
> **Request for a higher Rating  from Reviewer guSy**
>
> Dear Reviewer guSy,
>
> Thanks for taking the time to review and reply to our rebuttal. We are grateful for your feedback, which has helped us to improve our work. We understand and respect your decision to maintain your current rating. However, we kindly ask you to consider whether our clarifications justify a higher rating. We believe the enhancements made during the rebuttal, specifically the extended ablation studies for each component in our pipeline and both indoor and outdoor physical evaluations with previous methods,  have strengthened the quality and clarity of our work. We appreciate your understanding and consideration of this request. We would like to provide further clarification if there are any issues you would like us to address.
>
> Thanks again for your time and effort in reviewing our paper!
>
> Best Regards，
>
> Authors of Submission 17110

---

### Official Review · Reviewer_TBRf · 2024-07-17

**Soundness:** 4
**Presentation:** 3
**Contribution:** 4
**Rating:** 6
**Confidence:** 4

**Summary:**

The paper introduces a novel framework, CNCA, for generating customizable and natural adversarial camouflage for vehicle detectors using a diffusion model. This work addresses critical limitations in current adversarial camouflage techniques by focusing on naturalness and customizability, which are often neglected in favor of attack performance. While the paper presents a significant advancement in adversarial camouflage, several areas require improvement to enhance rigor and presentation. The proposed CNCA framework holds substantial promise, but further validation and detailed comparison are essential to establish its superiority and practical relevance.

**Strengths:**

1.	The use of a diffusion model for generating natural and customizable adversarial camouflage is novel.
2.	The extensive experiments, including both digital and physical settings, provide strong evidence of the method's effectiveness.

**Weaknesses:**

1.	The explanation of the adversarial feature generation and its integration with the diffusion model is somewhat convoluted. Quantitatively define the evaluation indicators of naturalness and attack performance, or provide relevant references.
2.	The evaluation in the physical world is limited to small-scale models and specific conditions. Extend the evaluation to a broader range of vehicle detection models and datasets, including those used in autonomous driving (e.g., KITTI, Waymo Open Dataset). Assess the scalability of CNCA by testing on larger, more complex scenes and different environmental conditions to validate its general applicability.
3.	The paper lacks ablation studies to isolate the impact of different components of the proposed framework. Conduct ablation studies to demonstrate the contribution of each component (e.g., the diffusion model, adversarial feature clipping) to the overall performance.

**Questions:**

Experimental Statistical Significance: The authors recruited 45 participants to subjectively evaluate the naturalness of different camouflages, reporting the mean scores and standard deviations (SD) for naturalness of each type of camouflage. While this is good, merely reporting the mean scores and SD does not statistically demonstrate whether the differences in mean scores are significant. It would be more convincing to conduct t-tests or ANOVA (preferably repeated measures ANOVA with post hoc tests, based on the current experimental design) and report the relevant statistics (e.g., t and F values, as well as p values).

Physical World Evaluation: In the physical world evaluation, the paper only compared two models, one for a normal and another for the generated camouflage. Have the authors considered including models with other adversarial camouflage methods for comparisons, as the authors did in the digital world?

**Limitations:**

Perform a thorough comparison with state-of-the-art methods like AdvCam and UAPs that are known for their effectiveness. Discuss the differences in performance metrics such as attack success rate, naturalness, and computational efficiency. Highlight the advantages and limitations of CNCA relative to these methods.

---

> ### Author Rebuttal · Authors · 2024-08-06
>
> **W1: The explanation of the adversarial feature generation and its integration with the diffusion model is convoluted. Quantitatively define the evaluation indicators of naturalness and attack performance or provide relevant references.**
>
> During the normal T2I diffusion model inference process, the text prompt is encoded to a text feature vector with the shape of (N, W) to guide the denoising process. N is the number of tokens, and W is the feature dimension of the token embedding. Our method introduces an adversarial feature vector with the shape of  (M, W). M is a hyperparameter that defines the size of adversarial information. During the adversarial camouflage generation, the clipping strategy regulates the adversarial feature vector. Then, it is concatenated with the text embedding feature to form the combined feature vector with the shape of (M+N, W) as input to the diffusion model. The diffusion generates vehicle texture, and the detector processes the camouflaged vehicle image. The adversarial feature is optimized by the detector's adversarial loss.
>
> The quantitative measurement of attack effectiveness used in our paper is car AP@0.5. This metric is computed based on the precision-and-recall curve for the car category. It's a popular performance metric for object detection, whose definition can be found in[1]. Previous state-of-the-art methods like DTA and ACTIVE have used this metric to evaluate their attack performance. The most relevant previous work on quantitative measuring of naturalness is S. Li et al. [2]. They collected vehicle images and human rating data to train a model to assess the naturalness automatically. We try to use their trained model in our case. However, we found a bias of low scoring towards vehicles with full-covered paintings, even if the painting is human-designed and looks natural. All the baselines except DAS receive low scores. We suspect it is because the dataset lacks full-cover painting vehicle images. As a result, we follow the previous work of Hu et al. [3] to conduct a subjective survey to evaluate the naturalness directly. To maintain the fairness of the survey, we invited 45 participants from different ages, backgrounds, and genders, which is twice the number (24) in [3].
>
> [1] Everingham, Mark, et al. "The Pascal visual object classes (VOC) challenge." International journal of computer vision 88 (2010): 303-338.
>
> [2] S. Li et al., “Towards Benchmarking and Assessing Visual Naturalness of Physical World Adversarial Attacks,” In Proceedings of the IEEE/CVF Conference on Computer Vision and Pattern Recognition. 2023, pp. 12324–12333, May 2023.
>
> [3] Hu, Yu-Chih-Tuan, et al. "Naturalistic physical adversarial patch for object detectors." Proceedings of the IEEE/CVF International Conference on Computer Vision. 2021.
>
> **W2: Assess the scalability of CNCA by testing on larger, more complex scenes and different environmental conditions to validate its general applicability.**
>
> The camouflage generated by our work needs to be deployed on a 3D model in the simulator or on a real vehicle model to get test data. Our generated camouflage cannot extend to the KITTI and Waymo datasets, which only contain 2D videos and images of vehicles. We try to extend our physical evaluation in indoor and outdoor scenarios and compare our method with previous baselines, as shown in Table 2 and Figure 1 in the attached PDF. The results show that our methods can achieve competitive attack performance among previous baselines. We acknowledge that our current evaluation is limited in scale and conditions due to our limited resources and budget. We agree that the road test evaluation for our method is essential, especially for autonomous driving. But due to its high time and labor cost, we plan to explore this in our future work.
>
> **W3: The paper lacks ablation studies to isolate the impact of different components of the proposed framework.**
>
> We have added extra ablation studies to demonstrate the contribution of each component, including the diffusion model, adversarial feature, clipping strategy, and mask reordering. Due to the word limits, please find the details of the ablation study in G1 section the global Author Rebuttal.
>
> **Q1: Experimental Statistical Significance: conduct t-test and ANOVA test to prove the significance of the naturalness survey data.**
>
> Thanks for your suggestion. We conduct a t-test and ANOVA with a post-hoc Tukey HSD test on our data. The results of the t-test are:
>
> | Compare Method | t Value | p Value |
> |----------------|---------|-----------|
> | Normal         | 8.72    | < .00001  |
> | DAS            | -3.18   | 0.00108   |
> | FCA            | -2.77   | 0.00358   |
> | DTA            | -4.04   | 0.00007   |
> | ACTIVE         | -4.13   | 0.00005   |
>
> The results of the ANOVA with post-hoc test are:
>
> | Compare Method | Q Value | p Value |
> |----------------|---------|---------|
> | Normal         | 11.18   | 0.001   |
> | DAS            | 4.77    | 0.011   |
> | FCA            | 4.44    | 0.023   |
> | DTA            | 5.91    | 0.001   |
> | ACTIVE         | 6.08    | 0.001   |
>
> The p values of the t-test and ANOVA test are lower than 0.05. Hence, we can conclude that the differences between the baselines and CNCA in the naturalness evaluation are significant.
>
>
> **Q2: Include models with other adversarial camouflage methods for comparisons in the physical world evaluation.**
>
> Please refer to the reply to W2.
>
> **L1: Compare CNCA with methods like AdvCam and UAPs.**
>
> Methods like AdvCam and UAPs can generate natural and effective physical adversarial examples based on one 2D image of the object. However, these physical adversarial examples are not robust against diverse viewing angles because they are typically optimized for a fixed viewing angle. As a result, these methods are not competent for our task because the attack needs to be effective at various viewing angles and distances. Therefore, we did not compare these methods with CNCA.

---

> > ### Comment · Reviewer_TBRf · 2024-08-14
> >
> > The authors provided thorough explanations and additional experiments to address concerns. The integration of adversarial features has been clarified with the diffusion model and provided relevant metrics for naturalness and attack performance. Additional studies were conducted, isolating the impact of each component in the framework. The authors extended their evaluation to more complex scenarios, though they acknowledge limitations due to resource constraints.
> >
> > The authors have mentioned that this paper need larger-scale evaluations in future work, with more immediate tests in varied environments.

---

> ### Author Response · Authors · 2024-08-14
> **Response to Reviewer TBRf**
>
> Dear Reviewer TBRf,
>
> We sincerely appreciate your time and effort in reviewing our work and rebuttal. We kindly ask you to **consider whether our improvements might justify a higher rating**. We believe the **extended ablation studies for each component in our pipeline and both indoor and outdoor physical evaluations with previous methods** have strengthened the quality of our work. We would be happy to provide further clarifications if you have further questions.
>
> Thanks again for your time and effort in reviewing our paper!
>
> Best Regards，
>
> Authors of Submission 17110

---

### Author Rebuttal · Authors · 2024-08-06

We want to thank all the reviewers for their insightful comments on our work. Most reviewers mention the ablation studies of CNCA pipeline components and comparisons with the previous baselines in the physical world. Hence, we have extended the ablation study and physical evaluation as suggested, which are discussed in the following sections.

**G1. Ablation studies for each CNCA pipeline component.**

| Pipeline                    | No Diff. | Diff. | Diff. + Adv. | Diff. + Adv. + Clip | Diff. + Adv. + Clip + Reorder |
|-----------------------------|----------|-------|--------------|---------------------|------------------------------|
| AP@0.5                 | 0.619    | 0.553 | 0.520        | 0.494               | 0.479                        |
| Natural Score                   | 1.00        | 3.37  | 1.71         | 2.75                | 3.33                         |


The above table shows the results of the ablation studies for each component of the pipeline. During the ablation study, we gradually add each component to see their contribution to attack performance and naturalness. All the test pipelines with the diffusion model use the same input text prompt: "yellow black graffiti." The description for each test pipeline is the following:

- **No Diff.** directly optimizes the texture image of the vehicle at a pixel level, resulting in an unnatural texture;

- **Diff.** introduces the diffusion model to generate the texture image compared to **No Diff.**, which improves the naturalness score;

- **Diff.+Adv.** introduces the adversarial feature compared to **Diff.**. This enables the texture image generation guided by the adversarial gradient from the detector. With this component, the attack performance is improved, but the naturalness is compromised;

- **Diff.+Adv.+Clip** introduces the clipping strategy compared. to **Diff.+Adv.**, which improves the naturalness;

- **Diff.+Adv.+Clip+Reorder** is our final pipeline, which uses a reordered texture map compared to **Diff.+Adv.+Clip**, which further improves the attack performance and naturalness.

To summarize, the ablation studies demonstrate that all the components of our pipeline contribute to improving the camouflage's attack performance and naturalness. Please refer to Table 1 in the attached PDF for the above table and the corresponding generated texture images.

**G2. Comparison with the previous baselines in the physical world.**

We implement four previous baseline camouflages in the physical world for comparison and add more physical world testing scenarios. To be specific, our latest physical experiments are conducted in both indoor and outdoor scenarios. In each scenario, we choose two distances and two elevation angles. We also choose 24 azimuth angles for the outdoor scenario and 27 azimuth angles for the indoor scenario. The physical test set contains 204 images for each method. In the attached PDF,  the comparison of the methods is shown in Table 2; examples of indoor and outdoor environments are shown in Figure 1. The results show that our method still achieves comparable attack performance with previous baselines in the physical world.

---

### Comment · Area_Chair_KAeb · 2024-08-11

Dear Reviewers,

The discussion period ends within 3 days. Authors made tremendous efforts on providing responses to your concerns. So, if you haven’t yet, please check responses and express your opinions whether you want to initiate discussion or accept the authors’ responses.


Best,
Your AC

---

### Decision · Program_Chairs · 2024-09-25

**Decision:**

Accept (poster)

**Comment:**

This paper introduces a natural way to generate adversarial camouflage using diffusion models, claiming that the proposed method is more natural than the closest method (ACTIVE [19]). Naturalness is achieved by using diffusion models and is evaluated via human evaluation.

This paper is on the borderline, but I all reviewers agreed on the novelty of this paper on the natural and customized adversarial camouflage via diffusion model. I think this paper could introduce a novel view in generating adversarial camouflage and this overweights the technical limitations. So, I vote for acceptance.

In your final manuscript, please address discussed reviewers’ concerns. Additionally,
* To mitigate the complexity of the algorithm, we recommend releasing code for the research community.
* Please discuss limitations related to focusing on vehicle detection. We agree that vehicle detection is an important task, but to my understanding, having UV maps and 3D mesh could introduce difficulties in using this attack. Please discuss this limitation in Section 5 if you agree.